# Dual-branch Robust Unlearnable Examples

**Xianlong Wang**[1] **Hangtao Zhang**[2] **Wenbo Pan**[1] **Ziqi Zhou**[3] **Changsong Jiang**[4] **Li Zeng**[5] **Xiaohua Jia**[1]

## Abstract

Unlearnable examples (UEs) aim to compromise model training by injecting imperceptible perturbations to clean samples. However, certain existing UE schemes exhibit limited robustness against advanced defenses due to their heuristic design or narrowly scoped domain perturbations. To address this, we propose DUNE, a **D**ual-branch **UN**learnable **E**nsemble perturbation optimization approach. Specifically, DUNE separately optimizes perturbations in the spatial and color domains to establish the mapping between perturbations and shift-induced labels. This design extends the perturbation domain to increase noise intensity for improving robustness and drives the models to learn perturbation-oriented features with degraded generalization, thereby achieving unlearnability. To strengthen DUNE's performance, we further propose an unlearnability-enhancing ensemble strategy that aggregates diverse pre-trained models during the dual-branch optimization. Extensive experiments on benchmark datasets CIFAR-10 and ImageNet verify that DUNE's robustness outperforms 12 SOTA UE schemes under 7 mainstream defenses, yielding a lower average test accuracy of 14.95% to 50.82%. The code is available at https://github.com/wxldragon/DUNE.

## 1. Introduction

With the flourishing development of *deep neural networks* (DNNs), a large number of machine learning platforms obtain public training data through web crawling (Carlini

---

[1]Department of Computer Science, City University of Hong Kong, Hong Kong SAR, China [2]University of Pennsylvania, Philadelphia, USA [3]Huazhong University of Science and Technology, Wuhan, China [4]University of Electronic Science and Technology of China, Chengdu, China [5]Changsha University of Science and Technology, Changsha, China. Correspondence to: Li Zeng <xtuzengli@163.com>.

*Proceedings of the 43rd International Conference on Machine Learning*, Seoul, South Korea. PMLR 306, 2026. Copyright 2026 by the author(s).

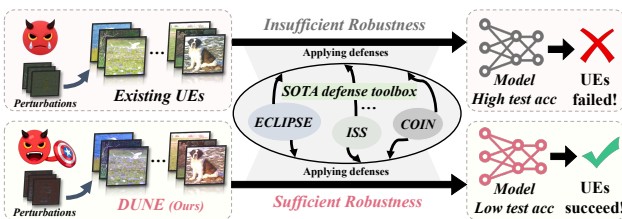

*Figure 1.* **Overview of UEs.** Existing UE schemes fail to compromise DNNs due to insufficient robustness, whereas our proposed scheme DUNE remains robust under these SOTA defenses.

et al., 2024; Baack, 2024). Given this, *unlearnable examples* (UEs) (Huang et al., 2021; Wang et al., 2024b; Yu et al., 2025; Sun et al., 2024; Wu et al., 2025) are proposed to make data "unlearnable" for deep learning models via adding imperceptible perturbations to clean images, thereby compromising model performance.

At a high level, UEs (Wu et al., 2023; Huang et al., 2021; Yu et al., 2022) exploit the tendency of DNNs to favor learning simpler shortcut (Geirhos et al., 2020; Wang et al., 2024c) features, establishing a mapping between perturbation features and labels for achieving unlearnability. To this end, existing UE schemes (Huang et al., 2021; Sandoval-Segura et al., 2022; Wu et al., 2023; Liu et al., 2024a; Sadasivan et al., 2023) typically search for *shortcut perturbations* easily captured by DNNs while ensuring perturbation stealthiness. However, such strategies primarily suffer from two limitations: (i) *Heuristic shortcut pattern.* Certain UEs (Yu et al., 2022; Sadasivan et al., 2023) tend to adopt empirical shortcut patterns directly as perturbations without formal optimization, rendering them easily compromised by adaptive defenses (Li et al., 2025; Liu et al., 2023); (ii) *Domain-constrained perturbation optimization.* Existing UE methods (Fu et al., 2022; Chen et al., 2023; Meng et al., 2024) further optimize perturbations to achieve unlearnability, using a single-domain constraint for preserving visual quality, which results in homogeneous perturbations vulnerable to noise-suppressing defenses (Wang et al., 2024a; Liu et al., 2023). Therefore, existing UE approaches exhibit insufficient robustness when confronted with the *state-of-the-art* (SOTA) defense toolbox, as demonstrated in Fig. 1. This motivates us to raise an intriguing research question:

*Is it possible to optimize shortcut perturbations across diverse domains while ensuring robustness?*

In response, we propose a **D**ual-branch **UN**learnable **E**nsemble scheme (`DUNE`) that optimizes perturbations in both the spatial and color domains for establishing a perturbation-to-label mapping. The orthogonality between these domains enriches perturbation diversity and thereby enhances robustness against various defenses. Specifically, we optimize perturbations in the feature space toward shift-induced labels that are displaced from the true labels, thereby forming a mapping from perturbations to shifted labels and achieving unlearnability. Following this principle, we decompose the joint dual-domain optimization into two independent sub-optimization problems, each optimizing perturbations within its respective domain. For spatial branch generation, we drive $\ell_\infty$-norm perturbations so that their features approach the shift-induced labels according to each sample's category, where the $\ell_\infty$ constraint guarantees perturbation stealthiness and the perturbation is optimized via $T$-step *project gradient descent* (PGD) (Madry et al., 2018). For color branch generation, our insight is to increase the orthogonality between color and spatial perturbations so as to reduce the overlap of shortcut patterns, thereby increasing perturbation diversity for improved robustness. To this end, the color perturbations are generated via shifting luminance corresponding to the *direct current* (DC) component, which is independent of the spatial perturbations in the *alternating current* (AC) components. In particular, distinct luminance-shifted unlearnable offsets are injected into each RGB channel, and a class-wise strategy with the gradient-free *particle swarm optimization* (PSO) (Wang et al., 2018) is employed to align them with the shift-induced labels. Finally, to further enhance `DUNE`'s unlearnability and robustness, we adopt the concept of ensemble learning (Dong et al., 2020), merging gradient or loss from a pre-trained model gallery during optimization to enrich the spectrum of perturbations, thus enhancing robustness.

Extensive experiments on benchmark datasets CIFAR-10 (Krizhevsky & Hinton, 2009) and ImageNet (Deng et al., 2009) verify that our proposed `DUNE` achieves superior robustness against 12 SOTA UE schemes under 7 strong defenses, accompanied by a lower average test accuracy of 14.95%∼50.82% on CIFAR-10. Beyond this, to evaluate `DUNE` under more challenging scenarios, we design two adaptive defenses (*i.e.*, assuming awareness of spatial-color domain knowledge) against `DUNE`, with experimental results on four diverse models confirming its robustness. Our main contributions are summarized as:

◆ We propose `DUNE`, a dual-domain unlearnable optimization scheme that generates spatial and color perturbations to align features with shift-induced labels, exhibiting sufficient robustness against defenses.

◆ We propose an unlearnability-enhancing ensemble module that aggregates gradient information from a

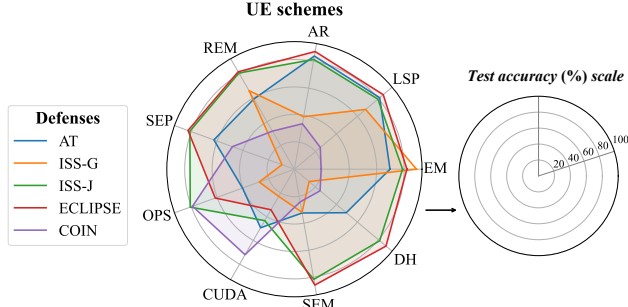

*Figure 2.* **Quantitative UE robustness** (in *test accuracy, %*) under five defenses with CIFAR-10 (Krizhevsky & Hinton, 2009) trained on ResNet18 (He et al., 2016). A point's proximity to the outer boundary signifies lower UE robustness, highlighting the limited robustness in existing UE schemes.

model gallery to enhance effectiveness and robustness.

◆ We conduct extensive experiments on multiple benchmark datasets and models, demonstrating that `DUNE` achieves superior robustness over 12 SOTA UE approaches under both existing and adaptive defenses.

## 2. Related Work

### 2.1. Unlearnable Examples

Unlearnable examples (UEs) introduce imperceptible perturbations to training samples, thereby degrading the generalization performance of unauthorized models. Existing UE schemes primarily generate perturbations under $\ell_p$-norm constraints in the spatial domain, such as $\ell_\infty$ methods *error-minimization* (EM) (Huang et al., 2021), *self-ensemble perturbation* (SEP) (Chen et al., 2023), and *stable error-minimization* (SEM) (Liu et al., 2024b), $\ell_2$ approaches *linearly separable perturbation* (LSP) (Yu et al., 2022) and *autoregressive perturbation* (AR) (Sandoval-Segura et al., 2022), and the $\ell_0$ scheme *one-pixel shortcut* (OPS) (Wu et al., 2023). Although these UEs achieve considerable unlearnability, their perturbations are constrained within an $\ell_p$-norm ball in the single spatial domain, rendering them vulnerable to noise-suppression defenses (Liu et al., 2023; Wang et al., 2024a; Tao et al., 2021) and thus lacking robustness. Another branch of UEs involves heuristic perturbation design, *e.g.*, convolutional noise CUDA (Sadasivan et al., 2023), which lacks robustness under the adaptive defense (Li et al., 2025). Thus, existing studies highlight the need for a more robust UE scheme.

### 2.2. Defenses Against UEs

Tao *et al.* (Tao et al., 2021) first employ *adversarial training* (AT) (Madry et al., 2018) defense to break unlearnability, but it is afterwards defeated by the *robust error-minimization* (REM) (Fu et al., 2022) UE scheme trained with adversarial perturbations. Subsequently, *orthogonal projection-based*

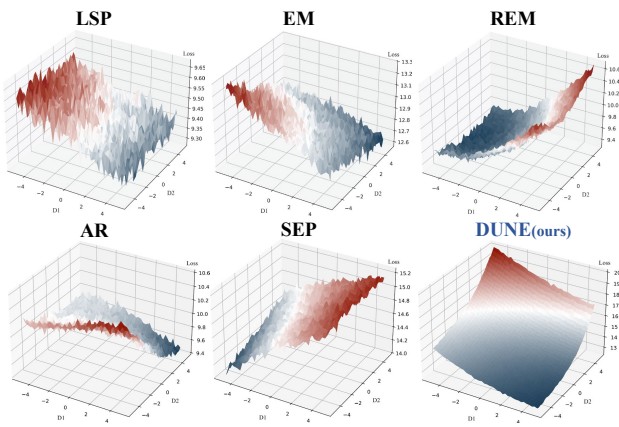

*Figure 3.* **Visualization of loss landscapes** for ResNet18 trained on five single-domain UEs (LSP (Yu et al., 2022), EM (Huang et al., 2021), REM (Fu et al., 2022), AR (Sandoval-Segura et al., 2022), SEP (Chen et al., 2023)) and our DUNE on CIFAR-10. $D_1$ and $D_2$ denote the normalized directions in the parameter space.

(OP) (Sandoval-Segura et al., 2023) and VAE-based (Yu et al., 2024a) approaches are proposed, enriching the defensive landscape. Beyond these defense methods, mainstream defenses include diffusion purification approaches (Wang et al., 2024a; Dolatabadi et al., 2024; Jiang et al., 2023) such as ECLIPSE (Wang et al., 2024a) and *image shortcut squeezing* (ISS) defense such as ISS-J (Liu et al., 2023), both of which can defeat existing single-domain UEs. Additionally, an adaptive defense COIN (Li et al., 2025) that employs random matrix transformation addresses convolutional UEs, further revealing the insufficient robustness of existing UEs, as quantitatively illustrated in Fig. 2.

## 3. Preliminaries

### 3.1. Problem Formulation

**UE's goal.** As data publishers intend to prevent the data from being exploited to train unauthorized DNNs $\mathcal{F}$, they integrate imperceptible noise $\delta_u$ to clean training samples $x \in \mathbb{R}^{C \times H \times W}$ to form the *unlearnable examples* (Huang et al., 2021; Fu et al., 2022; Zhu et al., 2024; Yu et al., 2024b), resulting in low generalization performance. Formally, the goal of UEs is to satisfy the following objectives:

$$\max_{\delta} \mathbb{E}_{(x',y') \sim \mathcal{D}_t} \mathcal{L}(\mathcal{F}(x'; \theta^*), y'), \quad (1)$$

$$\text{s.t. } \theta^* = \arg\min_{\theta} \sum_{(x,y) \in \mathcal{D}_c} \mathcal{L}(\mathcal{F}(x + \delta_u; \theta), y) \quad (2)$$

where $\delta_u$ refers to the perturbation applied to $x$ to serve as the unlearnable example $x_u = x + \delta_u$, $y$ denotes the ground-truth label, $(x', y')$ refers to data sampled from the clean testing distribution $\mathcal{D}_t$, $\mathcal{D}_c$ is clean training set, $\mathcal{L}$ represents the loss function, *e.g.*, cross-entropy loss, and $\theta^*$ denotes the optimized model parameters.

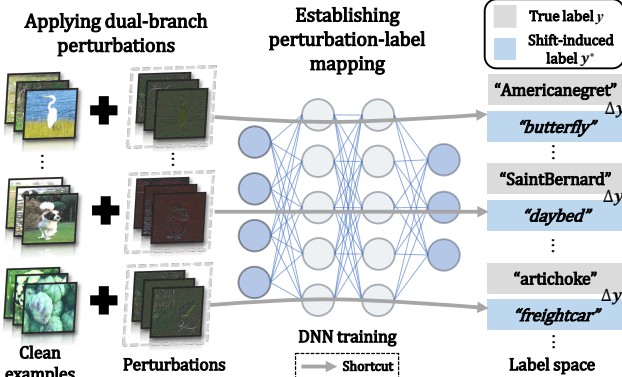

*Figure 4.* **An intuitive understanding** of DUNE's objective that pushes features toward the shift-induced class in a class-wise way.

Existing UE studies generate $\delta_u$ within a single domain, typically the spatial domain $\Phi_s$ instantiated by an $\ell_p$-norm ball space, *i.e.*, $\|\delta_u\|_p \leq \epsilon$. Such single-domain perturbations are easily compromised by existing defenses (Wang et al., 2024a), leading to the minimization of Eq. (1)'s expectation term, *i.e.*, UEs lack robustness against defenses.

**A typical UE design.** Beyond achieving UE objectives in Eqs. (1) and (2), several studies (Fu et al., 2022; Sadasivan et al., 2023; Meng et al., 2024) also increase the robustness of UEs against defenses, among which we select REM (Fu et al., 2022) as a representative. REM aims to minimize the adversarial loss by solving a min-min-max tri-level optimization problem to generate UEs, thereby enhancing robustness against AT (Tao et al., 2021). Specifically, the objective of generating perturbations is formulated as follow:

$$\min_{\delta_u} \mathbb{E}_{(x,y) \sim \mathcal{D}_c} \left[ \min_{\|\delta_u\|_\infty \leq \epsilon_u} \max_{\|\delta_a\|_\infty \leq \epsilon_a} \mathcal{L}(f_\theta^*(x + \delta_u + \delta_a), y) \right] \quad (3)$$

where $\delta_u$ denotes the unlearnable noise constrained within the norm range of $\epsilon_u$, and $\delta_a$ represents the $\epsilon_a$-bounded adversarial noise, $f_\theta^*$ denotes a randomly initialized network. REM optimizes Eq. (3) to drive $x_u = x + \delta_u$ toward class $y$ in feature space, thereby achieving unlearnability. Despite REM's robustness to AT due to tri-level optimization, restricting $\delta_u$ to a single $\ell_\infty$ domain confines the perturbation range, leaving it susceptible to stronger defenses (Wang et al., 2024a; Liu et al., 2023), as shown in Fig. 2.

### 3.2. Limitations of Existing Efforts

As previously discussed, existing UE schemes (Fu et al., 2022; Liu et al., 2024b; Meng et al., 2024; Wu et al., 2023) exhibit insufficient robustness against powerful defenses (Wang et al., 2024a; Liu et al., 2023), and this limitation can be attributed to two key factors: ❶ **Heuristic shortcut design.** Some existing UE approaches overly rely on empirically designed simple shortcut patterns, such as linear block perturbations (Yu et al., 2022) or convolutional noise (Sadasivan et al., 2023), without a principled opti-

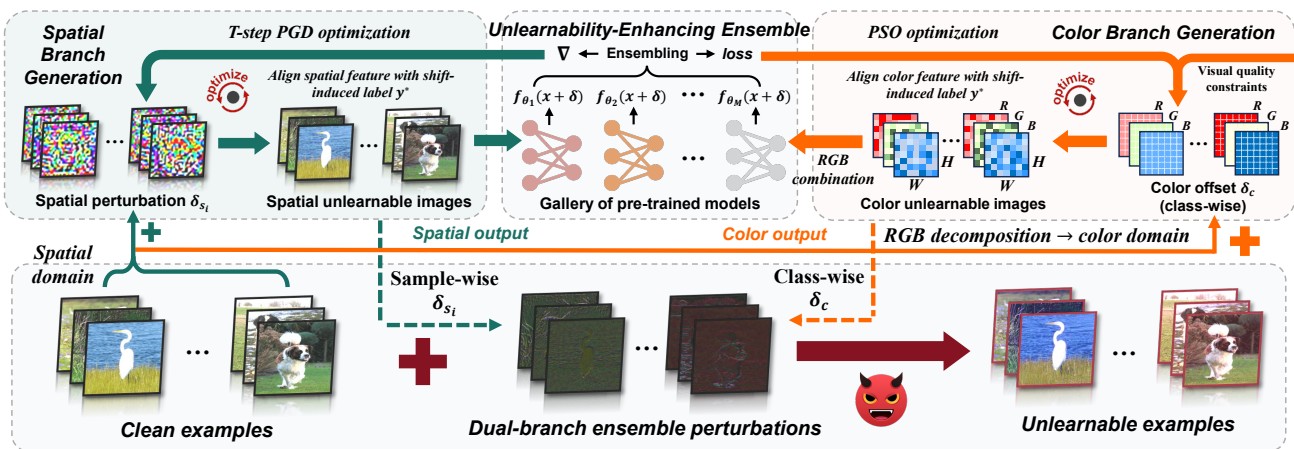

Figure 5. The framework of DUNE.

mization formulation. Such perturbations can be easily exploited by adaptive defenses (Li et al., 2025; Liu et al., 2023) to fully capture internal mechanisms and defeat them; ❷ **Domain-limited perturbation generation.** Existing UE schemes (Huang et al., 2021; Fu et al., 2022; Chen et al., 2023) optimize perturbations within a single domain, which confines them to a limited feature space and causes non-smooth and locally oscillating structures as shown in Fig. 3. Such loss landscapes rely on a single frequency pattern and are thereby fragile against defenses like ISS-J (Liu et al., 2023), which suppress frequency information. By comparison, DUNE yields a smoother landscape, highlighting the robustness of perturbations (Pham et al., 2024).

## 4. Methodology

### 4.1. Design Principle

To address the limitations of existing UEs, our design principles are listed as follows: ❶ **Feature misalignment optimization.** Given the fragility of heuristic designs, our principle is to increase the complexity of shortcut perturbations by minimizing the distance between features and shift-induced classes. Specifically, for each class, samples are optimized to approach the class shifted by the same label offset, as illustrated in Fig. 4. In this way, DNNs establish a mapping between perturbations and shift-induced labels, preventing them from learning sample features and reducing generalization ability; ❷ **Dual-branch perturbation design.** UEs (Wu et al., 2023) typically exploit shortcut perturbations (Geirhos et al., 2020) that contain simple features for DNN learning. Therefore, such shortcut noise may reside in any domain of image data, provided it remains easily learned by DNNs. Meanwhile, as complementary to the spatial domain, the color domain (Zhao et al., 2023; Wang et al., 2022) yields color perturbations that follow a fundamentally different distribution from the Gaussian noise purified in ECLIPSE (Wang et al., 2024a), while its low-frequency

nature enhances resilience to high-frequency compression defense ISS-J (Liu et al., 2023). Given these insights, our principle is to optimize perturbations in dual domains to increase noise diversity, thus boosting UE robustness.

### 4.2. A High-level Objective

**Pushing perturbations closer to shift-induced class.** Our idea is to generate shortcut perturbations in both the spatial domain $\Phi_s$ and the color domain $\Phi_c$ for establishing the mapping between dual-domain perturbation features and labels. Specifically, the perturbations are optimized toward the shift-induced classes $y^*$ in feature space, establishing a class-wise offset mapping as shown in Fig. 4. In this way, owing to their tendency to rely on shortcuts rather than intrinsic sample features, DNNs suffer from poor generalization, i.e., satisfying Eqs. (1) and (2), while perturbing across domains strengthens UE robustness. Formally, the high-level objective of DUNE is defined as:

$$\min_{\delta_u} \mathbb{E}_{(x,y)\sim\mathcal{D}_c}[\mathcal{L}_{\text{CE}}(f_\theta(\psi(x;\delta_u)), y^*)], \quad (4)$$

$$\text{s.t.} \quad \delta_u \in \Phi_s \times \Phi_c, \quad y^* = (y + \Delta y) \bmod k \quad (5)$$

where perturbation $\delta_u$ is derived in spatial-color domain, $f_\theta$ is a pre-trained network, $\mathcal{L}_{\text{CE}}$ is cross-entropy loss, $\psi$ denotes the operation of applying $\delta_u$ to sample $x$ to generate the UE, $y$ is the true label, $\Delta y$ indicates the label deviation (see Fig. 4), and $k$ refers to the number of categories in $\mathcal{D}_c$.

Grounded in the spatial-color perturbations, we analyze the robustness as presented below:

> **Remark I (*Frequency perspective*)**
>
> *Spatial perturbations concentrate on high-frequency components, enabling them to bypass low-frequency defenses such as ISS-G; color perturbations reside in low-frequency components, allowing them to evade defenses, e.g., ECLIPSE. This design forms a coupled cross-domain bias rather than a simple additive combination of two independent cues.*

## 4.3. A Ground-level Design: DUNE

**Unlearnable domain decomposition.** To reduce the complexity of dual-domain unlearnable objective in Eqs. (4) and (5), our key intuition is that both *independent* and *joint* dual-branch optimization exert negligible influence on the shift-induced mapping. From this perspective, we simplify the original joint optimization problem into two distinct domain-wise optimization problems (*i.e.*, $\delta_u \triangleq \delta_s \oplus \delta_c$) via decomposing the objective, each solved independently and re-formulated as follows:

$$\min_{\delta_d \in \Phi_d} \mathbb{E}_{(x,y)\sim\mathcal{D}_c}\big[\mathcal{L}_{\text{CE}}\big(f_\theta(\psi(x; \delta_d)), y^*\big)\big], d \in \{s, c\} \quad (6)$$

$$x_u = \psi(x; \delta_c, \delta_s) \quad (7)$$

where $(\delta_d, \Phi_d)$ corresponds to two domain-specific cases, each optimized separately by using Eq. (6) to obtain spatial perturbation $\delta_s$ and color perturbation $\delta_c$, which are then integrated through $\psi$ to produce $x_u$.

**Spatial branch optimization.** For spatial-domain unlearnable realization, we instantiate $\Phi_s$ as a domain formed by an $\ell_\infty$-norm ball sphere, formally expressed as: $\|\delta_s\|_\infty \leq \epsilon$, where $\epsilon$ is the norm constraint. Nevertheless, Eq. (6) for the spatial branch remains an intractable non-convex optimization. Given the perturbation is optimized in a class-wise deviation manner, we employ the $T$-step *project gradient descent* (PGD) (Madry et al., 2018) on the shift-induced label-based loss ranging from the classes. Assuming the clean dataset $\mathcal{D}_c$ is formulated as:

$$\mathcal{D}_c = \big\{ \underbrace{(x_1, y_p), (x_2, y_p), \ldots, (x_{n_p}, y_p)}_{n_p \text{ samples from class } y_p} \big\}_{p=1}^k \quad (8)$$

where $k$ is the number of classes. For each sample $x_i \in \{x_i\}_{i=1}^{n_p}$ belonging to ground-truth label $y_p$, we derive the perturbation $\delta_{s_i}$ in the spatial domain to align its feature with shift-induced class $y_p^*$ as follows:

$$g_t = \nabla_{x_i^t} \mathcal{L}_{\text{CE}}(f_\theta(x_i^t), y_p^*), \quad y_p^* = (y_p + \Delta y) \bmod k \quad (9)$$

$$x_i^{t+1} = x_i^t - \beta \cdot \text{sign}(g_t)\Big|_{t=0}^{T-1} \quad (10)$$

$$x_i^{t+1} = \min\big\{ \max\{x_i^{t+1}, x_i^0 - \epsilon\}, x_i^0 + \epsilon \big\}\Big|_{t=0}^{T-1} \quad (11)$$

where $x_i^0 = x_i$, $\delta_{s_i} = x_i^T - x_i^0$, $g_t$ denotes the gradient at the $t$-th iteration, serving to optimize the spatial perturbation $\delta_{s_i}$, $\beta$ is the step size, and $T$ is the number of iteration, after which the spatial branch generates spatial perturbation $\delta_{s_i}$. Eq. (11) ensures that $\delta_{s_i}$ is produced within the $\epsilon$-norm constraint, thus safeguarding sample visual quality.

**Color branch optimization.** *Motivation for luminance-shift perturbations.* An image can be decomposed into a *direct current* (DC) component representing block-wise mean

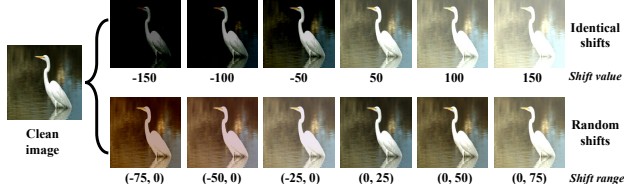

*Figure 6.* The luminance adjustment effect of applying identical pixel shifts and random pixel shifts to $R$, $G$, and $B$ channels.

luminance and *alternating current* (AC) components where spatial perturbations concentrate their energy (Guo et al., 2025). Therefore, to promote orthogonality between dual-domain perturbations for enhancing robustness, the color branch aims to optimize perturbations for altering luminance targeting the DC component. As shown in Fig. 6, applying either identical or distinct random shifts to RGB channels alters luminance, where the identical pattern includes only a single tunable parameter, restricting the optimization flexibility. Hence, we apply distinct shifts to each RGB channel, increasing the perturbation dimensionality. Formally, we define this process applied to $x_i$ as described below:

$$x_i \in \mathbb{R}^{3 \times H \times W} \xrightarrow{decompose} x_r, x_g, x_b \in \mathbb{R}^{1 \times H \times W}, \quad (12)$$

$$x_{u_i}^c = (x_r + \Delta x_r, x_g + \Delta x_g, x_b + \Delta x_b) \quad (13)$$

where $\Delta x_r$, $\Delta x_g$, and $\Delta x_b$ denote the luminance-shifted unlearnable offsets applied to the RGB decomposition $x_r$, $x_g$, and $x_b$, $x_{u_i}^c$ is the color-branch UE, and the perturbation $\delta_{c_i} = x_{u_i}^c - x_i$. For establishing the mapping between color-branch features and shift-induced labels, we assign class-wise perturbation $\delta_{c_i}$ to samples based on categories.

*Class-wise color-branch perturbation optimization.* As the optimization in Eqs. (6) and (13) for $\delta_{c_i}$ does not rely on gradients, we adopt the gradient-free *particle swarm optimization* (PSO) (Wang et al., 2018) that employs particles to navigate color space $\Phi_c$ to converge toward perturbation $\delta_{c_i}$ with a class-wise mode. Additionally, as shown in Fig. 6, similarity functions PSNR (Tanchenko, 2014), SSIM (Bakurov et al., 2022), and LPIPS (Kettunen et al., 2019) are employed to constrain the color domain $\Phi_c$. In light of these, from each class, we randomly select $N$ samples $\{x_i\}_{i=1}^N$ to construct their loss objective for PSO-driven minimization as follows:

$$\min_{\delta_{c_i}} \mathbb{E}_{\{(x_i, y_p)\}_{i=1}^N}[\underbrace{\mathcal{L}_{\text{CE}}(f_\theta(x_i + \delta_{c_i}), y_p^*)}_{unlearnability} + \underbrace{\lambda \mathcal{L}_{nc}(x_i, x_i + \delta_{c_i})}_{noise\ constraint}],$$

$$(14)$$

$$\text{s.t.,} \quad \delta_{c_1} = \delta_{c_2} = \cdots = \delta_{c_N} \equiv \delta_c \quad (15)$$

where $\delta_c$ denotes the same perturbation applied to samples $\{x_i\}_{i=1}^{n_p}$ from the same class $y_p$, whereas different classes employ different ones (*i.e.*, class-wise pattern), $\mathcal{L}_{nc} = \max(0, \tau_1 - \mathcal{L}_{psnr}) + \max(0, \tau_2 - \mathcal{L}_{ssim}) + \max(0, \mathcal{L}_{lpips} - \tau_3)$, and $\tau_1, \tau_2, \tau_3, \lambda$ are pre-defined hyper-parameters.

*Table 1.* **Evaluation and comparison.** The test accuracy (%) results on CIFAR-10 UE baselines across defenses with intra-architecture ResNet18 and cross-architecture VGG19 (since the surrogate model is ResNet18), where "AVG" indicates column-wise average values.

| Model | Defense | EM | TAP | URP | LSP | AR | REM | SEP | OPS | CUDA | GUE | SEM | DH | DUNE |
|---|---|---|---|---|---|---|---|---|---|---|---|---|---|---|
| ResNet18 | w/o | 18.26±0.86 | 31.41±5.41 | 17.89±1.36 | 22.73±1.51 | 11.68±1.00 | 25.81±1.36 | 9.42±1.72 | 25.32±2.66 | 25.48±2.21 | 11.71±0.66 | 15.94±2.06 | 10.28±0.49 | **13.26±1.12** |
| | AT | 69.72±1.22 | 82.81±0.37 | 84.72±1.07 | 81.19±0.65 | 83.80±0.63 | 59.12±2.40 | 62.51±1.23 | 40.28±1.65 | 49.32±2.16 | 77.32±0.77 | 32.43±1.12 | 49.37±2.27 | **24.96±0.63** |
| | AA | 82.08±4.35 | 66.46±1.41 | 90.23±0.50 | 85.93±2.46 | 45.83±5.62 | 43.34±3.13 | 22.79±2.60 | 46.55±18.02 | 40.78±1.04 | 31.28±16.76 | 39.29±5.14 | 46.06±3.42 | **19.55±3.18** |
| | OP | 64.37±5.09 | 46.35±0.55 | 89.39±0.38 | 89.73±0.39 | 29.45±3.98 | 30.18±0.92 | 12.58±3.71 | 88.98±1.02 | 28.66±2.48 | 88.33±0.46 | 15.99±3.05 | 82.67±8.23 | **12.81±2.52** |
| | ISS-G | 89.09±0.60 | 20.99±1.43 | 60.94±6.96 | 67.83±7.38 | 38.87±1.92 | 66.16±4.55 | 9.66±1.84 | 27.33±2.43 | 22.89±3.23 | 89.52±0.31 | 31.94±0.81 | 14.03±4.04 | **10.18±1.53** |
| | ISS-J | 78.91±0.52 | 81.32±0.65 | 81.09±0.62 | 79.50±0.08 | 81.33±0.23 | 80.99±0.58 | 81.12±0.21 | 80.87±0.05 | 43.31±1.61 | 79.34±0.03 | 81.58±0.53 | 81.19±0.78 | **28.88±2.22** |
| | ECLIPSE | 82.07±0.86 | 86.33±0.20 | 87.08±0.54 | 84.58±0.32 | 87.16±0.24 | 82.01±0.50 | 82.59±0.51 | 61.47±0.36 | 34.18±1.79 | 85.60±0.77 | 85.82±0.49 | 87.22±0.36 | **57.49±2.17** |
| | COIN | 19.49±1.15 | 78.24±0.52 | 81.56±1.20 | 24.68±2.42 | 33.67±3.04 | 32.07±2.02 | 48.10±0.70 | 79.09±1.35 | 72.02±0.21 | 18.82±3.36 | 24.22±4.72 | 24.49±1.95 | **19.21±1.65** |
| | AVG | 63.00±0.15 | 61.74±0.73 | 74.11±0.99 | 67.02±0.88 | 51.47±0.93 | 52.46±0.78 | 41.10±0.93 | 56.24±1.44 | 39.58±1.17 | 60.24±1.80 | 40.90±0.42 | 49.42±1.16 | **23.29±0.86** |
| VGG19 | w/o | 19.61±1.37 | 26.89±1.16 | 18.19±2.14 | 23.19±1.58 | 13.65±1.14 | 29.65±2.76 | 9.43±1.40 | 20.95±2.34 | 26.67±2.30 | 14.20±1.00 | 35.89±2.43 | 9.83±0.32 | **13.23±0.63** |
| | AT | 65.58±1.18 | 80.97±0.33 | 83.61±0.37 | 78.62±0.56 | 80.39±0.27 | 63.54±0.30 | 75.05±1.09 | 43.74±1.17 | 46.80±2.57 | 76.61±1.65 | 65.03±1.17 | 45.41±2.86 | **21.72±1.20** |
| | AA | 81.83±3.30 | 57.92±3.26 | 84.69±2.14 | 56.31±40.31 | 27.49±2.46 | 38.00±24.37 | 15.29±5.20 | 42.06±19.16 | 38.16±5.90 | 50.75±6.06 | 53.78±5.60 | 29.59±12.26 | **17.52±0.84** |
| | OP | 80.59±1.16 | 52.96±1.27 | 86.85±0.70 | 87.63±0.18 | 15.73±2.45 | 32.48±0.53 | 10.19±1.10 | 86.48±0.83 | 30.01±1.31 | 85.80±1.41 | 41.77±3.50 | 17.53±3.27 | **18.10±3.95** |
| | ISS-G | 86.40±0.53 | 26.18±1.47 | 62.44±2.22 | 81.74±0.95 | 37.76±2.03 | 70.96±1.44 | 7.26±3.11 | 19.24±2.13 | 19.52±3.40 | 88.16±0.68 | 44.67±0.93 | 10.35±0.17 | **10.00±1.32** |
| | ISS-J | 79.14±0.44 | 80.93±0.25 | 80.83±0.16 | 78.63±0.12 | 81.13±0.53 | 79.84±0.65 | 80.27±0.53 | 79.32±1.15 | 41.22±1.31 | 77.69±0.33 | 80.03±0.99 | 80.92±0.70 | **29.31±0.24** |
| | ECLIPSE | 78.78±1.69 | 84.59±0.36 | 85.27±0.73 | 83.85±0.33 | 85.21±1.18 | 80.47±0.50 | 80.59±0.47 | 64.64±1.61 | 32.37±0.32 | 83.63±0.80 | 80.27±8.43 | 85.67±0.80 | **56.38±1.26** |
| | COIN | 19.71±1.26 | 74.88±0.93 | 78.55±1.39 | 25.04±1.26 | 28.47±6.48 | 28.54±3.44 | 49.18±3.15 | 74.14±0.70 | 73.48±0.44 | 16.61±1.40 | 43.59±1.55 | 22.42±2.18 | **15.88±0.12** |
| | AVG | 63.95±0.40 | 60.66±0.41 | 72.55±0.74 | 64.38±5.24 | 46.23±1.13 | 52.93±3.03 | 40.91±0.97 | 53.82±2.66 | 38.53±1.81 | 61.68±1.31 | 55.63±0.65 | 37.72±1.72 | **22.77±0.37** |

Beyond the robustness gained from the dual-domain perturbations, this ensemble module provides an additional robustness boost (see Tab. 6), as analyzed below:

> **Remark II (*Loss landscape perspective*)**
>
> *Ensemble-based optimization enhances robustness by reducing model-specific biases and smoothing the loss landscape (see Fig. 3). By averaging gradients across models, it preserves only cross-model shortcut directions, yielding more stable perturbations against diverse defense approaches.*

**Unlearnability-enhancing ensemble.** To enhance the sample's unlearnability, we leverage the idea of ensemble learning (Dong et al., 2020) into both of the spatial branch and color branch UE generation process, where combined models guide the optimization direction. Assuming there exists a checkpoint gallery composed of pre-trained models $\{f_{\theta_i}\}_{i=1}^{M}$, Eq. (9) is reformulated to derive $g_t$ as below:

$$g_t = \frac{1}{M}\sum_{j=1}^{M}\nabla_{x_i^t}\mathcal{L}_{CE}(f_{\theta_j}(x_i^t), y_p^*) \quad (16)$$

Similarly, the cross-entropy loss $\mathcal{L}$ in Eq. (14) is replaced by $\frac{1}{M}\sum_{j=1}^{M}\mathcal{L}_{CE}(f_{\theta_j}(x_i+\delta_{c_i}), y_p^*)$, which is a gradient-free ensemble loss term. These two formulations revert to vanilla ones when $M = 1$, implying that ensemble models are not utilized and only a single model is used. For class $y_p$, after obtaining $\{\delta_{s_i}\}_{i=1}^{n_p}$ and $\delta_c$ from the spatial and color branches, we generate the UE $x_{u_i} = x_i + \delta_{s_i} + \delta_c|_{i=1}^{n_p}$. Therefore, the unlearnable dataset $\mathcal{D}_u$ is formulated as:

$$\mathcal{D}_u = \Big\{ \underbrace{(x_{u_1}, y_p), (x_{u_2}, y_p), \dots, (x_{u_{n_p}}, y_p)}_{n_p \text{ unlearnable examples from class } y_p} \Big\}_{p=1}^{k} \quad (17)$$

The framework of DUNE scheme is illustrated in Fig. 5.

*Table 2.* The test accuracy on ImageNet UEs trained on ResNet18.

| Defense ↓ UE → | EM | URP | LSP | REM | DUNE |
|---|---|---|---|---|---|
| w/o | 2.80±0.17 | 55.28±2.42 | 19.39±1.18 | 20.93±3.13 | 6.78±0.31 |
| AT (Tao et al., 2021) | 44.02±0.53 | 44.05±1.18 | 43.33±2.17 | 31.65±1.97 | **15.00±0.41** |
| AA (Qin et al., 2023) | 53.40±2.50 | 53.27±2.74 | 49.80±2.30 | 51.64±1.10 | **49.22±2.49** |
| ISS-G (Liu et al., 2023) | 41.72±5.73 | 44.20±9.33 | 50.17±2.27 | 41.55±0.41 | **14.67±3.02** |
| ISS-J (Liu et al., 2023) | 41.12±3.69 | 56.53±1.16 | 52.35±3.26 | 24.23±2.11 | **7.93±0.99** |
| COIN (Li et al., 2025) | 3.12±0.24 | 50.28±2.43 | 34.80±4.57 | 19.02±1.31 | **6.40±0.49** |
| AVG | 31.03±1.08 | 50.60±0.63 | 41.64±0.99 | 31.50±1.35 | **16.67±0.30** |

## 5. Experiments

### 5.1. Experimental Details

**Experimental setup.** Following existing UE works (Li et al., 2025; Meng et al., 2024; Wang et al., 2024a), we employ CIFAR-10 (Krizhevsky & Hinton, 2009) and ImageNet (Deng et al., 2009) (select the first 100 classes) datasets, along with ResNet18 (He et al., 2016), VGG19 (Simonyan & Zisserman, 2015), DenseNet121 (Huang et al., 2017), and EfficientNet-B0 (Tan & Le, 2019) networks. We employ SGD for 80 epochs training with a momentum of 0.9, an initial learning rate of 0.1, and a batch size of 128 for CIFAR-10 and 256 for ImageNet. Regarding the DUNE implementation, $f_\theta$ is set to ResNet18, and the hyper-parameters $M$, $\beta$, $T$, $N$, $\lambda$, $\Delta y$, $\epsilon$ are empirically set to 5, 0.5, 30, 1000, 1.0, 3, 8/255, respectively. We select 7 SOTA defenses with their default implementations, including AT (Tao et al., 2021), AA (Qin et al., 2023), OP (Sandoval-Segura et al., 2023), ISS-G (Liu et al., 2023), ISS-J (Liu et al., 2023), ECLIPSE (Wang et al., 2024a), and COIN (Li et al., 2025), to evaluate the robustness of UEs. Further experimental setup details are provided in Sec. A.

**Comparison baselines.** We compare the robustness of our proposed UE scheme DUNE with 12 SOTA UE methods, covering EM (Huang et al., 2021), TAP (Fowl et al., 2021), URP (Tao et al., 2021), LSP (Yu et al., 2022), AR (Sandoval-Segura et al., 2022), REM (Fu et al., 2022), SEP (Chen

*Table 3.* DUNE performance against ViT-based architectures.

| Datasets | CIFAR-10 | | ImageNet-100 | |
|---|---|---|---|---|
| Defenses | ViT-Tiny | ViT-Small | ViT-B/32 | ViT-B/16 |
| w/o | 13.32±2.21 | 13.01±2.72 | 2.37±0.47 | 2.53±0.38 |
| ISS-J | 18.12±0.69 | 16.53±0.74 | 3.42±0.28 | 2.88±0.25 |
| ISS-G | 23.07±9.88 | 25.88±6.91 | 10.73±0.78 | 8.13±0.62 |

*Table 4.* Computational cost of DUNE on different datasets.

| Datasets | Optimization | Per class | Per sample | Total time (hrs) |
|---|---|---|---|---|
| CIFAR-10 | PGD | 0.0346 | 6.92E-06 | 0.3459 |
| | PSO | 0.0957 | 1.92E-05 | 0.9574 |
| ImageNet-100 | PGD | 0.0136 | 6.80E-05 | 1.3641 |
| | PSO | 0.0758 | 3.79E-04 | 7.5766 |

et al., 2023), OPS (Wu et al., 2023), CUDA (Sadasivan et al., 2023), GUE (Liu et al., 2024a), SEM (Liu et al., 2024b), and DH (Meng et al., 2024). These selected UEs cover perturbation forms based on convolution, $\ell_0$, $\ell_2$, and $\ell_\infty$ norms, enabling a systematic robustness assessment and comparison of existing baseline approaches. The values in Tabs. 1 and 2 covered by deep cyan indicate the highest robustness, while those in light cyan represent the second-highest level of robustness.

**Evaluation metrics.** Consistent with previous UE literature (Meng et al., 2024; Li et al., 2025; Liu et al., 2024a; Huang et al., 2021), we evaluate UE performance by measuring the *test accuracy* (in %) of classifiers trained on unlearnable training datasets using clean test sets. Lower test accuracy indicates stronger unlearnability, while lower test accuracy under defense implies higher robustness of the UE. The test accuracy results of Tabs. 1 and 2 are reported as the average ± standard deviation across three runs using random seeds of 0, 1025, and 2025.

## 5.2. Evaluation and Comparison

**Effectiveness.** As shown in Tab. 1, without applying defense, DUNE reduces the test accuracy of ResNet18 and VGG19 to 13.26% and 13.23%, respectively, which is comparable to random guessing on the test set. Meanwhile, on the ImageNet100 in Tab. 2, DUNE also underscores the test accuracy to 6.78%, further confirming the effectiveness of the proposed DUNE scheme in achieving unlearnability. The results of ViT in Tab. 3 and retrieval task in Tab. 5 also reveal the effectiveness and robustness of DUNE.

**Robustness.** We evaluate the robustness of our proposed UE scheme DUNE under 7 diverse defense methods and further quantitatively compare its performance with 12 SOTA UE baselines, as demonstrated in Tabs. 1 and 2. Across diverse defenses, models, and datasets, DUNE consistently yields the lowest average test accuracy, reflecting the best overall robustness. Moreover, the highest accuracy achieved

*Table 5.* DUNE performance on the image-text retrieval task.

| Flickr8K dataset | Image-to-Text | | Text-to-Image | |
|---|---|---|---|---|
| Methods | Hit@10 (↓) | Medr (↑) | Hit@10 (↓) | Medr (↑) |
| Clean baseline | 22.7 | 58 | 18.5 | 60 |
| DUNE | 13.9 | 116 | 16.7 | 83 |
| DUNE + ISS-J | 15.6 | 114 | 15.8 | 99 |

*Table 6.* **Ablation study** on DUNE with CIFAR-10.

| Model | Num. | SB | CB | UEE | w/o | AT | ISS-G | ISS-J | COIN | AVG |
|---|---|---|---|---|---|---|---|---|---|---|
| Res-Net18 | ❶ | ✓ | | | 28.59 | 78.91 | 18.68 | 80.66 | 78.43 | 57.05 |
| | ❷ | | ✓ | | 32.41 | 42.47 | 70.93 | 47.25 | 25.29 | 43.67 |
| | ❸ | | ✓ | ✓ | 20.70 | 28.96 | 64.44 | 31.93 | 17.37 | 32.68 |
| | ❹ | ✓ | | ✓ | 12.77 | 77.50 | **9.82** | 80.65 | 70.26 | 50.20 |
| | ❺ | ✓ | ✓ | | 26.28 | 40.70 | 17.99 | 42.54 | 28.22 | 31.15 |
| | ❻ | ✓ | ✓ | ✓ | **12.54** | 24.41 | 11.85 | **31.18** | 18.37 | **19.67** |
| VGG-19 | ❶ | ✓ | | | 30.56 | 62.96 | 16.56 | 80.40 | 77.81 | 53.66 |
| | ❷ | | ✓ | | 29.67 | 33.61 | 79.28 | 40.37 | 25.46 | 41.68 |
| | ❸ | | ✓ | ✓ | 18.28 | 22.06 | 57.60 | **29.05** | 16.91 | 28.78 |
| | ❹ | ✓ | | ✓ | 16.10 | 66.92 | **10.35** | 80.61 | 61.87 | 47.17 |
| | ❺ | ✓ | ✓ | | 23.75 | 30.41 | 20.49 | 39.75 | 21.15 | 27.11 |
| | ❻ | ✓ | ✓ | ✓ | **13.67** | 20.59 | 10.89 | 29.30 | **15.98** | **18.09** |

by applying 7 SOTA defenses to DUNE remains below 60%, significantly lower than that of clean trained models, *i.e.*, 80%∼90%, indicating its superior robustness. As shown in Tab. 6, AT and ISS-J are primarily tailored to suppress high-frequency noise, and thus struggle against DUNE's color-branch perturbations; conversely, ISS-G targets low-frequency noise but remains ineffective against spatial-branch perturbations. Consequently, our dual-branch design yields robustness against these defenses.

**Time Overhead.** The results in Tab. 4 shows that DUNE introduces only modest computational overhead on both CIFAR-10 and ImageNet-100. Although PSO is consistently more time-consuming than PGD, the overall generation cost remains manageable, with total time below 1 hour on CIFAR-10 and below 8 hours on ImageNet-100 (200 samples per class). For ensemble model training, the per-forward FLOPs remain constant at 1.12G, since all ensemble members share the same architecture. In addition, the training times for the five checkpoints are 209.4 s, 610.4 s, 1185.9 s, 1371.9 s, and 1721.9 s, respectively. These results suggest that the additional overhead mainly comes from repeated forward evaluations of multiple checkpoints, rather than increased per-model complexity.

## 5.3. Ablation Study

To investigate the contribution of each component in DUNE, we conduct ablation studies, where the *spatial branch generation* module is denoted as SB, the *color branch generation* module as CB, and the *unlearnability-enhancing ensemble* module as UEE. As seen in Tab. 6, employing any single module or pair among SB, CB, and UEE yields lower average unlearnability performance across defenses compared with the DUNE scheme, confirming the critical contribution

*Table 7.* DUNE performance under joint defenses on CIFAR-10.

| Models | Defenses | EM | REM | TAP | LSP | AR | SEP | DUNE(Ours) |
|---|---|---|---|---|---|---|---|---|
| ResNet18 | Per-image channel normalization+ISS-J | $79.51_{\pm0.53}$ | $80.91_{\pm0.19}$ | $81.28_{\pm0.54}$ | $79.22_{\pm0.43}$ | $81.89_{\pm0.37}$ | $80.83_{\pm0.61}$ | $\mathbf{67.97_{\pm0.87}}$ |
| | LAB whitening+ISS-J | $79.33_{\pm0.51}$ | $81.28_{\pm0.41}$ | $80.98_{\pm0.42}$ | $79.31_{\pm0.66}$ | $81.36_{\pm0.51}$ | $80.18_{\pm0.95}$ | $\mathbf{56.69_{\pm1.67}}$ |
| VGG19 | Per-image channel normalization+ISS-J | $78.44_{\pm0.32}$ | $80.72_{\pm0.28}$ | $80.53_{\pm0.51}$ | $79.25_{\pm0.65}$ | $80.68_{\pm0.84}$ | $80.36_{\pm0.50}$ | $\mathbf{65.77_{\pm0.86}}$ |
| | LAB whitening+ISS-J | $79.09_{\pm0.31}$ | $80.63_{\pm0.27}$ | $80.12_{\pm0.98}$ | $78.49_{\pm1.32}$ | $80.38_{\pm1.06}$ | $80.13_{\pm0.41}$ | $\mathbf{56.05_{\pm1.40}}$ |

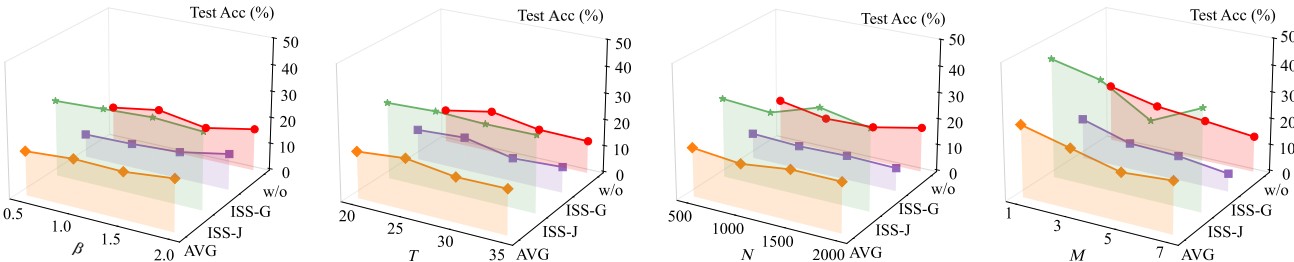

*Figure 7.* **Hyper-parameter sensitivity analysis.** The impact of hyperparameters $\beta$, $T$, $N$, and $M$ on the test accuracy results (%) of the ResNet18 classifier trained on DUNE-generated CIFAR-10 dataset.

of each module. Meanwhile, it is observed that removing the UEE module (using only SB and CB), *i.e.*, optimizing perturbations with a single model, causes the unlearnability performance to decrease by 13.74% on ResNet18 and 10.08% on VGG19, with robustness performance (under AT (Tao et al., 2021), ISS (Liu et al., 2023), COIN (Li et al., 2025)) also reduced by 5.17%-16.29%. These results indicate that UEE module plays a crucial role in improving the effectiveness and robustness of DUNE's perturbations.

### 5.4. Hyper-parameter Analysis

We select four hyper-parameters, $\beta$, $T$, $N$, and $M$, each of which plays a critical role in different modules of DUNE. As shown in Fig. 7, the $\beta$ and $T$ of the spatial branch achieve the best average performance at 0.5 and 30, respectively. The overall trend remains stable, except that excessively large $\beta$ degrades unlearnability. We attribute this to the overly large step hindering convergence in the spatial domain and weakening feature displacement. For the color branch parameter $N$, the best performance is achieved at $N = 1000$, where optimal unlearnability offsets in the color domain are obtained. Regarding the model ensemble parameter $M$, excessively small values reduce gradient diversity, while overly large values increase optimization complexity, both of which hinder overall unlearnability.

### 5.5. Resistance to Potential Adaptive Defenses

**Formulation of adaptive defenses.** Beyond standard settings, we also consider a more challenging threat model in which the defense is assumed to possess knowledge of DUNE. Given that DUNE produces class-wise feature separation across the spatial and color domains, we design an adaptive defense that introduces random noise in both

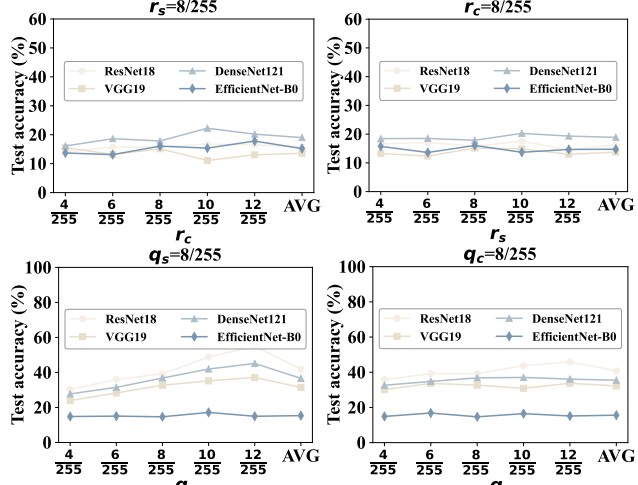

*Figure 8.* **Adaptive defenses.** DUNE performance under the two types of adaptive defenses on CIFAR-10.

domains to neutralize UEs, formulated as follow:

$$\forall x_u \sim \mathcal{D}_u, \quad x_u' \leftarrow x_u + \eta_c + \eta_s \quad (18)$$

where $\eta_c$ denotes channel-wise random shifts, independently sampled for each RGB channel from $\mathcal{U}(-r_c, r_c)$, $\eta_s$ represents spatial Gaussian noise sampled from $\mathcal{U}(0, r_s)$, and $r_c$ and $r_s$ respectively control the noise strength.

We further consider an adaptive defense, which substitutes random Gaussian noise with adversarial noise through adversarial training, while $\eta_c$ remains, formulated as:

$$\forall x_u \sim \mathcal{D}_u, \quad x_u' \leftarrow x_u + \eta_c, \quad (19)$$

$$\min_\theta \mathbb{E}_{(x_u',y)\sim\mathcal{D}_u}\left[ \max_{\|\eta_a\|\le q_s} \mathcal{L}_{CE}(\mathcal{F}(x_u' + \eta_a; \theta), y) \right] \quad (20)$$

where $\eta_c \sim \mathcal{U}(-q_c, q_c)$, $\eta_a$ is the adversarial noise during AT, and $q_c$ and $q_s$ control the strengths of the two noises.

**Resistance of DUNE.** The robustness of `DUNE` is evaluated under varying adaptive defense strengths across 4 model architectures, as demonstrated in Fig. 8. For the adaptive defense in Eq. (18), with either $r_c$ or $r_s$ held fixed, varying the random noise injected into another domain keeps the test accuracy of diverse models trained on `DUNE` data below 30%. For the adaptive defense in Eqs. (19) and (20), despite achieving better results than dual-branch random noise, the overall performance is relatively limited, further verifying the robustness of `DUNE` against such adaptive defenses. The robustness arises from that `DUNE` induces class-wise feature deviations across dual domains, thereby inducing representation-level biases that is hard to be neutralized by noise, consistent with findings in Wang *et al.*'s work (Wang et al., 2024b). Additionally, we evaluate `DUNE` against other common adaptive joint defenses in Tab. 7, the results consistently demonstrate the superior robustness of `DUNE` over the SOTA UE baselines.

## 6. Conclusion, Limitation, and Future Work

In this research, we propose `DUNE`, a dual-branch unlearnable example generation framework that jointly exploits spatial and color domain perturbations and enhances robustness through ensemble optimization. Unlike prior single-domain approaches, `DUNE` expands the unlearnable space while preserving semantic consistency, making it resistant to existing SOTA UE defenses. Extensive experiments on benchmark datasets demonstrate significant robustness improvements over existing UE methods. One current limitation is the main focus on image classification tasks, future research will extend `DUNE` to multi-modal or real-world tasks.

## Acknowledgements

This work was supported by the Sichuan Science and Technology Program under Grant 2026NSFSC1464 and Hong Kong Research Grants Council under Grant RFS2425-1S01.

## Impact Statement

This work studies robust unlearnable examples that preserve human-perceived utility while making data difficult for unauthorized models to learn. Such UEs can help protect privacy and intellectual property for individuals and data owners by reducing non-consensual model training and potential downstream misuse. They may also support data-governance policies in shared-data settings without requiring changes to model architectures or training pipelines.

However, the same capability could be misused to degrade legitimate model training, disrupt public datasets, or hinder research and safety efforts, acting as a denial-of-service vector against machine-learning development. To mitigate these risks, we encourage clear licensing and consent, provenance tracking, dataset integrity checks, robust detection and filtering, and responsible evaluation protocols. Overall, robust UEs promote user agency and data protection, but require careful governance to prevent adversarial abuse.

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

---

**Algorithm 1** PSO-based color-branch optimization

---

**Input:** Loss function $\mathcal{L}_{color}$; dimension $d$; number of particles $P$; maximum iterations $T_p$.
**Output:** Optimal color-branch perturbation $\delta_c$.
**Initialization:** Initialize particle positions $pos_j \sim \mathcal{U}(-0.1, 0.1)^d$, velocities $vel_j \sim \mathcal{U}(-0.01, 0.01)^d$;
Set personal best $pbest_j = pos_j$, $pbest\_val_j = +\infty$;
Set global best $gbest = \mathbf{0}$, $gbest\_val = +\infty$;
**for** $t = 1$ *to* $T_p$ **do**
   **for** $j = 1$ *to* $P$ **do**
      /* Evaluate loss for current particle */
      $cur\_val = \mathcal{L}_{color}(pos_j)$;
      /* Update personal best */
      **if** $cur\_val < pbest\_val_j$ **then**
         $pbest\_val_j = cur\_val$;
         $pbest_j = pos_j$;
      **end**
      /* Update global best */
      **if** $cur\_val < gbest\_val$ **then**
         $gbest\_val = cur\_val$;
         $gbest = pos_j$;
      **end**
   **end**
   /* Update particle velocity and position */
   $vel_j = 0.5 \cdot vel_j + 0.3 \cdot (pbest_j - pos_j) + 0.3 \cdot (gbest - pos_j)$;
   $pos_j = pos_j + vel_j$;
**end**
**Return:** Color-branch perturbation $\delta_c = gbest$.

---

## Appendix: Dual-branch Robust Unlearnable Examples

## A. Experimental Setup

Beyond the experimental details presented in the main paper, we provide the following supplementary notes: ImageNet inputs are uniformly cropped to $224 \times 224$. The hyper-parameters $\tau_1$, $\tau_2$, $\tau_3$ are empirically set to 20, 0.80, 0.03 respectively. The ensemble model checkpoints are derived from five distinct training epochs of a pre-trained ResNet18. The PSO (Wang et al., 2018) optimization of the color-branch perturbation is shown in Algorithm 1, where the hyperparameters $P$ and $T_p$ are empirically set to 10 and 10, respectively. During training, random flipping and random cropping are also employed.

## B. Comparison Baselines

In this section, we present the implementation details of the UE baselines used for comparison as follows:

- EM (Huang et al., 2021): We follow the official repository;

- TAP (Fowl et al., 2021): We follow the official implementation from the original paper;

- URP (Tao et al., 2021): We follow the official repository;

- LSP (Yu et al., 2022): We follow the official repository;

- AR (Sandoval-Segura et al., 2022): We follow the official repository;

- REM (Fu et al., 2022): We follow the official repository;

- SEP (Chen et al., 2023): We follow the official repository;

- OPS (Wu et al., 2023): We follow the official repository;

*Table 8.* DUNE performance against other SOTA defenses on CIFAR-10.

| Methods | ResNet18 | | | VGG19 | | |
|---|---|---|---|---|---|---|
| | AVATAR (Dolatabadi et al., 2024) | DVAE (Yu et al., 2024a) | ANSDA (Zhu et al., 2024) | AVATAR (Dolatabadi et al., 2024) | DVAE (Yu et al., 2024a) | ANSDA (Zhu et al., 2024) |
| EM | $82.15_{\pm0.25}$ | $75.44_{\pm1.15}$ | $83.55_{\pm0.11}$ | $81.78_{\pm1.03}$ | $77.41_{\pm2.73}$ | $81.08_{\pm0.81}$ |
| REM | $87.16_{\pm0.39}$ | $70.85_{\pm0.84}$ | $74.25_{\pm0.61}$ | $85.19_{\pm0.70}$ | $73.63_{\pm0.89}$ | $75.78_{\pm0.27}$ |
| LSP | $85.62_{\pm0.60}$ | $72.93_{\pm1.28}$ | $78.79_{\pm1.41}$ | $84.90_{\pm0.02}$ | $77.61_{\pm0.28}$ | $81.42_{\pm0.22}$ |
| DUNE (Ours) | $\mathbf{26.73_{\pm3.03}}$ | $\mathbf{39.84_{\pm2.49}}$ | $\mathbf{11.49_{\pm0.37}}$ | $\mathbf{24.68_{\pm0.07}}$ | $\mathbf{35.83_{\pm3.98}}$ | $\mathbf{14.96_{\pm0.41}}$ |

*Table 9.* Comparison of DUNE performance with spatial noise combined with Region-k.

| UEs | Models | w/o defense | ISS-J |
|---|---|---|---|
| Spatial branch + R4 | ResNet18 | $29.81_{\pm6.02}$ | $80.97_{\pm1.04}$ |
| Spatial branch + R4 | VGG19 | $19.57_{\pm3.48}$ | $79.75_{\pm0.09}$ |
| Spatial branch + R16 | ResNet18 | $32.16_{\pm1.41}$ | $81.20_{\pm0.88}$ |
| Spatial branch + R16 | VGG19 | $15.89_{\pm1.92}$ | $79.54_{\pm0.31}$ |
| DUNE (Ours) | ResNet18 | $\mathbf{13.26_{\pm1.12}}$ | $\mathbf{28.88_{\pm2.22}}$ |
| DUNE (Ours) | VGG19 | $\mathbf{13.23_{\pm0.63}}$ | $\mathbf{29.31_{\pm0.24}}$ |

- CUDA (Sadasivan et al., 2023): We follow the official repository;

- GUE (Liu et al., 2024a): We follow the official repository;

- SEM (Liu et al., 2024b): We follow the official repository;

- DH (Meng et al., 2024): We follow the official implementation from the original paper.

For fair comparison, we use the best hyper-parameters from their official implementations to generate the UEs by default. In addition, we supplement the experimental results under three SOTA defenses as shown in Tab. 8. These results also highlight the performance of DUNE.

Also, Tab. 9 shows that simply adding a low-frequency perturbation cannot reproduce the robustness of DUNE, especially under ISS-J. It shows that the benefit of DUNE should be better understood as arising from an orthogonal dual-domain design: while the spatial branch mainly perturbs AC-dominant spatial structures, the color branch directly alters image luminance by applying channel-wise RGB shifts that target the DC component. In this sense, the color branch is not simply another low-frequency spatial perturbation, but a complementary luminance-oriented perturbation in color space.

## C. Defense Implementations

In this section, we present the implementation details of the defense schemes used for evaluation as follows:

- AT (Tao et al., 2021): We follow the official repository;

- AA (Qin et al., 2023): We follow the official repository;

- OP (Sandoval-Segura et al., 2023): We follow the official repository;

- ISS-G (Liu et al., 2023): We follow the official repository;

- ISS-J (Liu et al., 2023): We follow the official repository;

- ECLIPSE (Wang et al., 2024a): We follow the official repository;

- COIN (Li et al., 2025): We follow the official repository.

By default, we adopt the optimal hyper-parameters provided in their official implementations.

