# OpenReview forum: "Dual-branch Robust Unlearnable Examples"
_ICML.cc/2026/Conference — ICML 2026 regular_

### Official Review · Reviewer_feDc · 2026-03-04

**Soundness:** 4
**Presentation:** 2
**Significance:** 3
**Originality:** 3
**Overall Recommendation:** 5
**Confidence:** 5

**Summary:**

This paper proposes a novel Dual-branch Unlearnable Ensemble perturbation optimization to generate effective unlearnable examples by separately optimizing perturbations in both spatial and color domains. They further strengthen their unlearnability by ensemble strategy that aggregates diverse pre-trained models during the dual-branch optimization. Experimental results demonstrate the superior performance under various UE defense paradigms.

**Compliance With Llm Reviewing Policy:**

Affirmed.

**Final Justification:**

My concerns have been fully solved. I'd like to increase my rating to 5 to support the acceptance of this paper.

**Key Questions For Authors:**

Questions:

1.	What do D1 and D2 represent in Figure 3?

2.	You said unlearnable perturbations in color domain may be close to low-frequency perturbations. I am wondering if low-frequency perturbations (together with normal spatial perturbations) on spatial domain can also obtain similar property with your DUNE? Because some existing UEs like Region-k [4] are low-frequent obviously.


Ref:

[4] Sandoval-Segura et al. Poisons that are learned faster are more effective. In CVPR 2022.

**Limitations:**

Yes.

**Strengths And Weaknesses:**

Strengths:

1.	The dual-branch optimization in spatial and color domains seems novel for UE generation.

2.	Experimental evaluation shows superior performance of proposed DUNE.

3.	Evaluation under custom adaptive defenses verifies generalization to challenging scenarios.

Weaknesses:

1.	The claim that “existing UE approaches are heuristic shortcut designs” is not accurate. In fact, [1] has proved that some UEs are linearly separable. Therefore, linear separability might be a fundamental explanation of UEs’ applicability.

2.	It seems that $\delta_c$ generated from color-branch doesn’t exist norm restriction. Therefore, this property makes the poison generator stronger as most of existing UEs are based on norm restriction (like $l_\infty=8$).

3.	More UE defense paradigms like ANSDA [1], AVATAR [2], DVAE [3] should be evaluated.

4.	Ensemble uses M=5 models may raise potential computational overheads. Reported metrics on runtime or resource consumption are lacked.

5.	Some typos exist. For instance, In Line 247, the citation of PGD doesn’t exist.

Ref:

[1] Zhu et al. Detection and defense of unlearnable examples. In AAAI 2024.

[2] Dolatabadi et al. The devil’s advocate: Shattering the illusion of unexploitable data using diffusion models. In IEEE SaTML 2024.

[3] Yu et al. Purify unlearnable examples via rate-constrained variational autoencoders. In ICML 2024.

---

> ### Author Rebuttal · Authors · 2026-03-31
>
> #### **Response to Reviewer feDc**
>
> Thank you for recognizing the novelty of our dual-branch design, the strong empirical performance, and its robustness under adaptive settings. We will respectfully address your constructive comments as below (Due to the space limit, some experimental results are provided through an external link. Thank you for your understanding):
>
> > **Q1:** "Heuristic shortcut" claim is not accurate.
>
> **Re:** We thank the reviewer for this important comment, and we fully agree that some UEs are not heuristic designs. In the revised version, we will rephrase it as: “certain existing UE schemes exhibit limited robustness against advanced defenses due to their heuristic design”, and we will also cite the relevant reference [1] in the main text to support this point. We again thank the reviewer for this professional and valuable suggestion.
>
> [1] Zhu et al. Detection and defense of unlearnable examples. AAAI 2024.
>
> ----------------
>
> > **Q2:** Color branch lacks norm constraint.
>
> **Re:** This is an important point, and we thank the reviewer for bringing it up. We acknowledge that the color branch does not impose a norm-based constraint, which may raise a fairness concern compared with prior norm-bounded UE methods. We kindly note that the color branch is still constrained through a visual similarity loss in Eq. (14), which preserves perceptual fidelity and avoids noticeable visual distortion. In this sense, our color branch still holds stealthiness constraint, but adopts a different form of constraint for the color-domain perturbation. We will clarify this point in the revised manuscript in Sec. 5 based on your insightful suggestions.
>
> -------------
>
> > **Q3:** More defenses should be evaluated.
>
> **Re:** We greatly appreciate this constructive feedback. To follow your suggestion, we additionally report the defense results in terms of test accuracy (%) on CIFAR-10 at📍[**here-are-exp-results**](https://anonymous.4open.science/r/More-defenses-62BA/).
>
> It can be seen that DUNE maintains accuracy below 40%, while prior methods remain above 70%. This confirms DUNE's superior robustness. We will include these results in Sec. 6 in the revised paper.
>
> ----------------
>
> > **Q4:** Ensemble incurs extra runtime cost.
>
> **Re:** We highly appreciate this helpful and constructive comment. We agree that using an ensemble introduces additional computational overhead, and that the practical cost should be reported more clearly. Following your suggestion, we report results as follow: the per-forward FLOPs remain constant at 1.12G, since all ensemble members share the same architecture. In addition, the training times for the five checkpoints are 209.4 s, 610.4 s, 1185.9 s, 1371.9 s, and 1721.9 s, respectively. The GPU (NVIDIA GeForce RTX 3090) memory usage is generally stable at around 2.9–3.1 GB, with average CPU usage ranging from approximately 16% to 36%. These results suggest that the additional overhead mainly comes from repeated forward evaluations of multiple checkpoints, rather than increased per-model complexity. We will include this analysis in the revised manuscript for a clearer discussion in Sec. 6.
>
> -------------
>
> > **Q5:** Typos and missing citations.
>
> **Re:** Thank you for pointing this out. We will correct the error in Line 247 and carefully check the manuscript for similar typographical issues in the revised version.
>
> -----------
>
> > **Q6:** Figure 3 symbols D1/D2 are unclear.
>
> **Re:** Thank you for noting this. In the revised manuscript, we will add an explicit explanation that D1 and D2 denote the two normalized directions in the model parameter space used to construct the plane for loss landscape visualization.
>
> -----------
>
> > **Q7:** Low-frequency perturbations with normal spatial perturbations need comparison.
>
> **Re:** This is a valuable comment, and we thank the reviewer for highlighting it. Following your suggestion, we compare DUNE with our spatial noise combined with Region-k (k=4 and 16). The results are at📍[**here-are-exp-results**](https://anonymous.4open.science/r/Spatial-RegionsK-1310/README.md).
>
> These results show that simply adding a low-frequency perturbation cannot reproduce the robustness of DUNE, especially under ISS-J. It shows that the benefit of DUNE should be better understood as arising from an orthogonal dual-domain design: while the spatial branch mainly perturbs AC-dominant spatial structures, the color branch directly alters image luminance by applying channel-wise RGB shifts that target the DC component (mentioned in Lines 271-274). In this sense, the color branch is not simply another low-frequency spatial perturbation, but a complementary luminance-oriented perturbation in color space.  We will strengthen this more accurate discussion in Sec. 4.3 and revise Remark I accordingly to reflect the above interpretation more precisely.
>
> We sincerely appreciate your careful review and valuable comments. Your feedback is very helpful to our revision.

---

> > ### Author Rebuttal · Reviewer_feDc · 2026-03-31
> >
> > Thanks for your rebuttal. My concerns have been fully solved. I'd like to increase my rating to 5 to support the acceptance of this paper.

---

> > > ### Author Response · Authors · 2026-04-01
> > >
> > > We sincerely thank you for your careful review and for taking the time to read our rebuttal. We are very grateful that our response has adequately addressed your concerns, as well as your kind support for the acceptance of this paper.

---

### Official Review · Reviewer_L7zV · 2026-03-08

**Soundness:** 3
**Presentation:** 3
**Significance:** 3
**Originality:** 3
**Overall Recommendation:** 5
**Confidence:** 4

**Summary:**

DUNE (Dual-branch UNlearnable Ensemble) is a robust unlearnable example (UE) generation framework that protects data privacy and intellectual property by modifying clean samples with imperceptible perturbations so that they are “unexploitable” to a malicious deep learning model. However, existing UE schemes are vulnerable to advanced defense toolboxes (e.g., image compression or diffusion purification) due to their ubiquitous heuristic designs and narrow perturbation domains. To this end, we propose DUNE to further improve the robustness of UE examples by learning the perturbations in both the spatial and color domains.

**Compliance With Llm Reviewing Policy:**

Affirmed.

**Key Questions For Authors:**

see weakness

**Limitations:**

Yes.

**Strengths And Weaknesses:**

A. Strengths:
DUNE tackles the intrinsic limitations of typical UE schemes via a well-motivated dual-branch design that simultaneously learns spatial (high-frequency) and color (low-frequency) domain perturbations for full-spectrum coverage to varied defenses. The orthogonality between branches inherently advances perturbation diversity and curtails shortcut pattern overlap for resistance even against adaptation threat models. In addition, the UEE module further enhances cross-architecture transferability via multi-model gradient aggregation to alleviate model-specific bias. Extensive experiments over different datasets/architectures and across 7 mainstream defenses, together with loss landscape visualizations, empirically and interpretable support for the efficacy of our DUNE.

B. Weaknesses:
a. Basic errors in prepositions, literature citations, articles, subject-verb agreement, etc.
1. Lines 91-92: The transitive verb ''lack'' can directly follow its object to express ''lack of something'' without requiring the preposition ''of''.
2. Lines 247-248: The literature citation regarding project gradient descent (PGD) failed to link successfully.
3. Lines 376-377: The phrase ''without applying defense'' lacks an article.
4. Lines 422-423: The use of the linking verb ''forming'' in ''forming representation-level biases that is hard to...'' is incorrect.

b. Optimization Complexity. Color branch relies on gradient-free PSO optimiziation, which can result in much higher computational overhead than the standard single-domain PGD. However there isn't a separate time cost analysis study, so it's hard to judge how scalable DUNE is in practice.

c. Task Limitation. DUNE only being tested on image classification benchmarks (CIFAR-10 and ImageNet), is not truly verified for generalizability to other real world domains. More experiments on more specific datasets, or even very early results on more challenging tasks such as multi-modal or real time video, would greatly improve the paper's practical relevance. Is it possible to conduct experiments on other datasets to briefly demonstrate DUNE's experimental results?

---

> ### Author Rebuttal · Authors · 2026-03-31
>
> #### **Response to Reviewer L7zV**
>
> Thank you very much for recognizing the strong motivation, technical novelty, and extensive validation of our approach. We will also respectfully address your insightful comments:
>
> > **Q1:** Writing and citation errors.
>
> **Re:** We sincerely thank the reviewer for these careful and helpful corrections. We will revise each of the issues you pointed out one by one in the revised manuscript. We have also proofread the manuscript again to further improve its overall writing quality.
>
> ---------
>
> > **Q2:** PSO lacks time-cost analysis.
>
> **Re:** We truly appreciate this thoughtful comment. We agree that the PSO-based color branch introduces higher complexity than PGD, and a dedicated runtime analysis is necessary. To address this concern, we report time cost results as:
>
> |   Dataset    |    CIFAR-10     |     CIFAR-10     | CIFAR-10 |  ImageNet-100   |   ImageNet-100   | ImageNet-100 |
> | :----------: | :-------------: | :--------------: | :------: | :-------------: | :--------------: | :----------: |
> | Time (hours) | Per class (AVG) | Per sample (AVG) |  Total   | Per class (AVG) | Per sample (AVG) |    Total     |
> |     PGD      |     0.0346      |     6.92E-06     |  0.3459  |     0.0136      |     6.80E-05     |    1.3641    |
> |     PSO      |     0.0957      |     1.92E-05     |  0.9574  |     0.0758      |     3.79E-04     |    7.5766    |
>
> These results verify that PSO is slower than PGD. We also gently note that this additional cost is incurred only during the offline poison-generation stage (enabling a richer search space that contributes to DUNE’s robustness), rather than during model training or inference. We will add this time-cost analysis in Sec. 6 based on your insightful suggestions.
>
> --------
>
> > **Q3:** Evaluation task and dataset are limited.
>
> **Re:** This suggestion is both important and helpful, and we are grateful for it. Following your suggestion, we evaluate DUNE on the multimodal retrieval task over the Flickr8K dataset [1] (since retrieval does not use explicit labels, we follow standard practice and use a surrogate model ResNet18 to generate pseudo-labels for DUNE) . The results are as follows:
>
> | Flickr8K       | I2T Hit@10 (↓) | I2T Medr (↑) | T2I Hit@10 (↓) | T2I Medr (↑) |
> | -------------- | -------------- | ------------ | -------------- | ------------ |
> | Clean baseline | 22.7           | 58           | 18.5           | 60           |
> | DUNE           | 13.9           | 116          | 16.7           | 83           |
> | DUNE+ISS-J     | 15.6           | 114          | 15.8           | 99           |
>
> These results demonstrate the unlearnability and robustness of DUNE in the multimodal retrieval setting. We will include this evaluation in Sec. 6 of the revised manuscript based on your advice.
>
> [1] [Framing image description as a ranking task: Data, models and evaluation metrics](https://www.jair.org/index.php/jair/article/view/10833). Journal of Artificial Intelligence Research
>
> We sincerely thank you for your careful review and thoughtful comments. Your feedback is very helpful for strengthening the paper.

---

> > ### Author Rebuttal · Reviewer_L7zV · 2026-04-01
> >
> > My concerns have been adequately addressed.

---

> > > ### Author Response · Authors · 2026-04-01
> > >
> > > We truly appreciate your thoughtful review and the time you devoted to considering our rebuttal. It is very encouraging to know that our clarification has satisfactorily addressed your concerns, and we are grateful for your positive reassessment of our paper.

---

### Official Review · Reviewer_8bxj · 2026-03-13

**Soundness:** 3
**Presentation:** 3
**Significance:** 2
**Originality:** 2
**Overall Recommendation:** 3
**Confidence:** 4

**Summary:**

DUNE proposes a dual‑branch approach to unlearnable examples by independently optimizing spatial PGD noise and class‑wise color perturbations, each targeting complementary frequency domains to confuse models and drive features toward shifted labels, with the main novelty being the introduction of a low-frequency (color) perturbation branch. The method uses ensemble models and a PSO‑based color branch, which works well for CIFAR‑10 and ImageNet‑100. Overall, the framework boosts robustness over prior UEs.

**Compliance With Llm Reviewing Policy:**

Affirmed.

**Key Questions For Authors:**

yes

**Limitations:**

How would your color‑branch hold up against simple, targeted pre‑processing (e.g., per‑image channel normalization, histogram equalization, or LAB/YUV whitening) that explicitly removes class‑wise luminance biases, given the perturbation is just three global channel offsets?

**Strengths And Weaknesses:**

**Strengths**

The dual‑branch design is conceptually interesting and does address known weaknesses of single‑domain UEs.

Evaluation shows clear improvements over prior methods under the standard defense suite.

**Weaknesses**

The color‑branch robustness feels superficial, since it relies on a simple class‑wise luminance shift that works mainly because current defenses don't target that direction; once color‑space normalization or similar preprocessing is considered, this advantage seems more like a blind‑spot exploitation of current defense assumptions than a fundamentally resilient mechanism.

The overall robustness appears compositional (i.e., combining perturbations that evade different defenses) rather than intrinsically hard to remove.

May not hold under jointly designed defenses targeting both domains (for example defenses that include per‑image channel normalization, histogram equalization, or LAB/YUV whitening)

The PSO‑based per‑class color optimization is computationally expensive and doesn't scale well to datasets with many classes.

The experimental setup relies only on older CNN architectures for both generation and evaluation, leaving generalization to modern models (e.g., transformers) untested, these newer models may rely less on shortcut features.

---

> ### Author Rebuttal · Authors · 2026-03-31
>
> #### **Response to Reviewer 8bxj**
>
> Thank you very much for recognizing the conceptual interest, practicality, and performance of our approach. We will also respectfully address the insightful concerns you have raised (Due to the space limit, some experimental results are provided through an external link. Thank you for your understanding):
>
> > **Q1:** Color branch is not resilient and works mainly because current defenses don't target that direction.
>
> **Re:** Thank you for this thoughtful comment. We agree that the color branch alone can be affected by targeted preprocessing. We respectfully feel that this concern may stem from viewing the color branch in isolation, whereas the robustness of DUNE is intended to be understood at the level of the full jointly optimized framework. Specifically, in Tab. 1 of our paper, we evaluated DUNE under color-targeted defenses (e.g., ISS-G that suppresses color noise by converting RGB images to grayscale, thereby removing channel-wise perturbations), test accuracy (%) results are as:
>
> | Using ISS-G defense |         ResNet18          |           VGG19           |
> | :-----------------: | :-----------------------: | :-----------------------: |
> |     DUNE(ours)      | $10.18\scriptsize{±1.53}$ | $10.00\scriptsize{±1.32}$ |
>
> It can be seen that DUNE remains robust against color-targeted defenses. We agree that this point can be clarified further, and we will refine the discussion in Sec. 4 based on your insightful suggestions.
>
> --------------------------
>
> > **Q2:** Robustness is compositional, not intrinsic.
>
> **Re:** Thank you for raising this insightful point about the mechanism. We agree whole robustness cannot be assumed from individual-domain robustness alone. We gently note that DUNE’s robustness indeed arises from joint cross-domain optimization, rather than combining separate perturbations. The supporting empirical results are shown in Tab. 1 and Fig. 8 of our paper: DUNE remains robust not only under ECLIPSE (targeting on both spatial and color noise) but also under the adaptive joint defenses that explicitly target both domains. In our view, this is because the spatial and color perturbations are jointly optimized toward the same shift-induced label (see Sec. 4.2), thereby forming a coupled cross-domain bias rather than a simple additive combination of two independent cues.  We will revise the discussion in Sec. 4 to make this point clearer. We sincerely thank the reviewer again for this insightful feedback.
>
> ------------
>
> > **Q3:** May fail under joint cross-domain defenses.
>
> **Re:** Thank you for proposing this constructive perspective. Following your helpful suggestions, we evaluate the color-targeted defenses you suggest and joint defenses with results at📍[**here-are-exp-results**](https://anonymous.4open.science/r/Joint-defenses-A2AF/README.md).
>
> These results show that jointly designed cross-domain defenses are indeed more effective than using color-targeted preprocessing alone, especially when combined with ISS-J. At the same time, they still do not neutralize DUNE, as the resulting test accuracy remains substantially below the clean-training level. We will incorporate these results in Sec. 6 in the revision and thank the reviewer for this constructive suggestion.
>
> ------------
>
> > **Q4:** PSO is costly and unscalable.
>
> **Re:** This is an insightful suggestion, and we sincerely appreciate it. We agree that the PSO per-class color optimization introduces additional computational overhead, and that its scalability may become less favorable as the number of classes grows. We kindly note that this cost is incurred only once in the perturbation-generation stage, rather than repeatedly during model training or inference. Moreover, the optimization is carried out in a low-dimensional space (three channel-offset values) rather than full image pixel space. For datasets with many classes, we agree that scalability is a limitation of the current design. We are very grateful to the reviewer for identifying this limitation, and we will discuss it explicitly in Sec. 6 of the revised manuscript.
>
> ---------
>
> > **Q5:** No validation on models like ViTs.
>
> **Re:** We sincerely thank the reviewer for this constructive comment. Following your suggestion, we added new experiments on ViT models for both DUNE-crafted CIFAR-10 and ImageNet-100 at 📍[**here-are-exp-results**](https://anonymous.4open.science/r/ViT-8694/).
>
> The results show that DUNE remains effective and robust on ViT models. We will include these results in Sec. 5 of the revised manuscript.
>
> Thank you very much for your thoughtful review and valuable feedback. We sincerely appreciate your time and comments, which are very helpful for improving the manuscript.

---

> > ### Author Rebuttal · Reviewer_8bxj · 2026-04-05
> >
> > The answer to Q3 confirms that while a simple joint defense doesn't neutralize DUNE it makes it much less effective. This makes DUNE's advantage over other methods less clear. PSO concern acknowledged but not addressed. I will keep my score.

---

> > > ### Author Response · Authors · 2026-04-07
> > >
> > > We sincerely appreciate your careful reading and your continued feedback, especially for pointing out the remaining two issues. Here are our clarifications:
> > >
> > > 🔴**Comparison of DUNE with existing schemes under joint defenses.**
> > >
> > > Following your feedback, we supplemented with direct comparisons between DUNE and UE baselines under the same joint cross-domain defenses on CIFAR10: Per-image channel normalization+ISS-J, LAB whitening+ISS-J, and the published work ECLIPSE [1], as:
> > >
> > > | ResNet18                              | EM                        | REM                       | TAP                       | LSP                       | AR                        | SEP                       | DUNE(Ours)                                                 |
> > > | ------------------------------------- | ------------------------- | ------------------------- | ------------------------- | ------------------------- | ------------------------- | ------------------------- | ---------------------------------------------------------- |
> > > | Per‑image channel normalization+ISS-J | $79.51\scriptsize{±0.53}$ | $80.91\scriptsize{±0.19}$ | $81.28\scriptsize{±0.54}$ | $79.22\scriptsize{±0.43}$ | $81.89\scriptsize{±0.37}$ | $80.83\scriptsize{±0.61}$ | $\textcolor{red}{\boldsymbol{67.97\scriptsize{\pm 0.87}}}$ |
> > > | LAB whitening+ISS-J                   | $79.33\scriptsize{±0.51}$ | $81.28\scriptsize{±0.41}$ | $80.98\scriptsize{±0.42}$ | $79.31\scriptsize{±0.66}$ | $81.36\scriptsize{±0.51}$ | $80.18\scriptsize{±0.95}$ | $\textcolor{red}{\boldsymbol{56.69\scriptsize{\pm 1.67}}}$ |
> > > | ECLIPSE(ESORICS'24)                   | $82.07\scriptsize{±0.86}$ | $82.05\scriptsize{±1.03}$ | $86.33\scriptsize{±0.20}$ | $84.58\scriptsize{±0.32}$ | $87.16\scriptsize{±0.24}$ | $82.59\scriptsize{±0.51}$ | $\textcolor{red}{\boldsymbol{57.49\scriptsize{\pm 2.17}}}$ |
> > > | **VGG19**                             | **EM**                    | **REM**                   | **TAP**                   | **LSP**                   | **AR**                    | **SEP**                   | **DUNE(Ours)**                                             |
> > > | Per‑image channel normalization+ISS-J | $78.44\scriptsize{±0.32}$ | $80.72\scriptsize{±0.28}$ | $80.53\scriptsize{±0.51}$ | $79.25\scriptsize{±0.65}$ | $80.68\scriptsize{±0.84}$ | $80.36\scriptsize{±0.50}$ | $\textcolor{red}{\boldsymbol{65.77\scriptsize{\pm 0.86}}}$ |
> > > | LAB whitening+ISS-J                   | $79.09\scriptsize{±0.31}$ | $80.63\scriptsize{±0.27}$ | $80.12\scriptsize{±0.98}$ | $78.49\scriptsize{±1.32}$ | $80.38\scriptsize{±1.06}$ | $80.13\scriptsize{±0.41}$ | $\textcolor{red}{\boldsymbol{56.05\scriptsize{\pm 1.40}}}$ |
> > > | ECLIPSE(ESORICS'24)                   | $78.78\scriptsize{±1.69}$ | $80.47\scriptsize{±0.50}$ | $84.59\scriptsize{±0.36}$ | $83.85\scriptsize{±0.33}$ | $85.21\scriptsize{±1.18}$ | $80.59\scriptsize{±0.47}$ | $\textcolor{red}{\boldsymbol{56.38\scriptsize{\pm 1.26}}}$ |
> > >
> > > (P.s.: ECLIPSE combines diffusion denoising against spatial noise with grayscale transformation against color noise, and thus serves as a strong SOTA joint cross-domain defense, with comparative results on DUNE and the other baselines being provided in Table1 of the original manuscript)
> > >
> > > These results show that, **DUNE still remains the most robust among the compared methods**, consistently yielding the lowest test accuracy and preserving a 12–25% absolute gap over the baselines under these joint defenses. Therefore, the cross-domain joint defenses do **not** remove DUNE’s relative advantage over prior approaches. We will include these results in the revised manuscript to make this clear.
> > >
> > > [1] ECLIPSE:Expunging Clean-label Indiscriminate Poisons via Sparse Diffusion Purification.ESORICS'24
> > >
> > > ---------------
> > >
> > > 🔴**PSO overhead.**
> > >
> > > Following your comment, we further provide the runtime cost of PSO on 10-class and 100-class datasets:
> > >
> > > |   Dataset   |    CIFAR-10    |    CIFAR-10     | CIFAR-10 |  ImageNet-100  |  ImageNet-100   | ImageNet-100 |
> > > | :---------: | :------------: | :-------------: | :------: | :------------: | :-------------: | :----------: |
> > > | Time(hours) | Per class(AVG) | Per sample(AVG) |  Total   | Per class(AVG) | Per sample(AVG) |    Total     |
> > > |     PSO     |     0.0957     |    1.92E-05     |  0.9574  |     0.0758     |    3.79E-04     |    7.5766    |
> > >
> > > These results suggest that, although the cost increases with the number of classes, the overhead remains practically manageable in the benchmark settings considered here, especially since it is incurred only once during the poison-generation stage rather than during model training or inference. We will also clarify this point of the revised paper.
> > >
> > > We are truly grateful for your constructive feedback, which helped us make these two points much more concrete. We respectfully hope that our revisions have addressed the remaining concerns, and we would be very grateful if the reviewer could reconsider the assessment in light of these new results.

---

### Decision · Program_Chairs · 2026-04-30

**Decision:**

Accept (regular)

**Comment:**

This paper proposes to optimize unlearnable perturbations in the spatial and color domains, an idea motivated by existing defenses ISS and ECLIPSE. The proposed method has shown very strong results. Two of the three reviewers gave a score of 5. The rest reviewer did provide initial construction comments, but did not submit the final justification. The AC has carefully checked the authors' response to that reviewer and found that the remaining concerns have been well addressed. Specifically, the reviewer states that "The answer to Q3 confirms that while a simple joint defense doesn't neutralize DUNE it makes it much less effective. This makes DUNE's advantage over other methods less clear." For this, the authors have supplemented extensive comparisons to show that the superiority of DUNE consistently holds. In addition, the authors provide detailed computational cost results of PSO.